# Additive Digital Casting: From Lab to Industry

**DOI:** 10.3390/ma15103468

**Published:** 2022-05-11

**Authors:** Ena Lloret-Fritschi, Elia Quadranti, Fabio Scotto, Lukas Fuhrimann, Thibault Demoulin, Sara Mantellato, Lukas Unteregger, Joris Burger, Rafael G. Pileggi, Fabio Gramazio, Matthias Kohler, Robert J. Flatt

**Affiliations:** 1Group of Physical Chemistry of Building Materials, Institute for Building Materials, 8093 Zurich, Switzerland; fabio.scotto1@gmail.com (F.S.); l.fuhrimann@bluewin.ch (L.F.); thibault.demoulin@gmail.com (T.D.); sara.mantellato@ifb.baug.ethz.ch (S.M.); flattr@ethz.ch (R.J.F.); 2Institute for Technology in Architecture, Gramazio Kohler Research, 8093 Zurich, Switzerland; quadranti.e@gmail.com (E.Q.); burger@arch.ethz.ch (J.B.); gramazio@arch.ethz.ch (F.G.); kohler@arch.ethz.ch (M.K.); 3Digital Fabrication in Architecture, Academy of Architecture, Università della Svizzera Italiana, 6850 Mendrisio, Switzerland; 4SACAC AG, Fabriksstrasse 11, 5600 Lenzburg, Switzerland; unteregger@sacac.ch; 5Department of Construction Engineering, University of São Paolo, São Paolo 01000-000, Brazil; rafael.pileggi@lme.pcc.usp.br

**Keywords:** processing, concrete, formwork, robotics, set on demand, digital concrete

## Abstract

Concrete construction harms our environment, making it urgent to develop new methods for building with less materials. Structurally efficient shapes are, however, often expensive to produce, because they require non-standard formworks, thus, standard structures, which use more material than is often needed, remain cheaper. Digital fabrication has the potential to change this paradigm. One method is Digital Casting Systems (DCS), where the hydration of self-compacting concrete is controlled on the fly during production, shortening the required setting time and reducing hydrostatic pressure on the formwork to a minimum. This enables a productivity increase for standard concrete production. More importantly, though, it enables a rethinking of formworks, as the process requires only cheap thin formworks, thus, unlocking the possibility to produce optimised structural members with less bulk material and lower environmental cost. While DCS has already proven effective in building structural members, this process faces the challenge of moving into industry. This paper covers the next steps in doing so. First, we present the benchmark and expectations set by the industry. Second, we consider how we comply with these requirements and convert our fast-setting self-compacting mortar mix into a coarser one. Third, we present the next generation of our digital processing system, which moves closer to the industrial requirements in terms of size and the control system. Finally, two prototypes demonstrate how DSC: (a) increases standard bulk production by 50% and (b) can be cast into ultra-thin formworks. We discuss the results and the short-term industrial concerns for efficiency and robustness, which must be addressed for such a system to be fully implemented in industry.

## 1. Introduction

In the prefab industry, concrete is traditionally shaped by casting it into rigid and heavy-duty standard formworks. After 8 to 24 h, these formworks are removed and prepared for reuse. It is a highly efficient process that, nevertheless, still has margin for improvement. In particular, the setting and hardening rates of self-compacting concrete can be optimized to allow either for a faster demolding or for the use of alternative formworks at low cost that offer a larger range of geometrical freedom. In the first case, having a self-compacting concrete that sets and hardens more rapidly allows the production rate to be increased, which is particularly valuable in accommodating variations in market demand. While various accelerating admixtures can be used for this purpose, they are often associated with a rapid flow loss, which can be problematic in practice. Thus, concrete must be activated just before placing, while also ensuring enough open time for casting and, nevertheless, benefiting from a rapid strength gain shortly thereafter.

Such systems may be referred to as setting on demand [1,2,3,4,5,6,7] and have recently received a lot of interest in the context of digital fabrication. Such systems involve preparing concrete with a retarder, to extend its open time, and adding an accelerator just before use. This is best executed in a continuous process, where the retarded concrete is pumped through a mixing reactor, where it is mixed with the activator before casting or extrusion. Such processes are central to rapid 3D printing [4,8,9], but also essential to a family of processes, termed Digital Casting Systems (DCS), in which heavy-duty formworks are then able to be replaced by millimeter-thin ones.

Using thin formworks reduces the price of elements of non-standard shapes that can often result from structural optimization, but they also present a solution for using less material in construction. Notably, the construction sector consumes 40–50% of the world’s natural resources and generates 30% of Europe’s waste [10,11,12], so there is a true need to facilitate the cost-effective production of elements that consume less material for a given load-bearing requirement. Thus, the objective of this paper is to highlight recent progress in developing digital concrete casting for an industrial pre-fab plant, with methods that combine the self-compacting qualities of concrete with rapid setting and hardening.

First, the paper presents a methodology, through which material developments at the laboratory scale may be streamlined to successful scale up from a concrete lab to comply with industrial requirements. Second, we present the next generation of our digital processing system, which both in terms of size and control system targets the processing of a coarser concrete mix, with aggregates up to 8 mm. Third, we demonstrate that our DCS can be used to remove standard formworks 50% faster compared to standard production processes in the concrete industry, and more importantly, that DCS enables the use of millimeter-thin formworks, which unlocks the possibility to produce structurally optimised shapes at low environmental cost. Finally, we discuss these results, as well as the short-term industrial concerns for efficiency and robustness that remain to be addressed for such a system to be fully implemented in industry.

## 2. Background and Requirements

### 2.1. Upscaling Concrete Processes

An important challenge when upscaling concrete mixes from laboratory to industry is to address the significant differences in not just the mixing energy (when using mixers with different volume capacity and configuration) but also in the mixing time, admixture addition, temperature changes, variation in water content caused by remains of water in the mixing equipment, humidity in the facilities and the storage condition of the material components, which is often outside. All of these factors can significantly impact both the rheological behavior and the hydration kinetics. Figure 1 shows an example of the slump flow evolution of a given concrete mix design, prepared with the same material mix in the academic lab and in industry. The concrete batch, produced in small volume in the lab, shows a much faster workability loss. The main differences between the production of these two batches are described in the Appendix A. In addition to the influences described above, we suspect that the water content has a significant impact on this difference. Boscaro et al. [13] proposed, for example, using the microwave method to double check the water content of industrially prepared concretes.

Such differences in moving to large-scale production can be accentuated in the presence of chemical admixtures, raising issues for setting-on-demand systems. It is, thus, an objective of this paper to lay out a simple procedure, allowing for the efficient upscaling of such concrete formulations to an industrial context.

The methodology presented includes various mix design optimization steps using cement pastes and mortars, which raises similar issues of mixing energy that must be accounted for. Setting-on-demand systems combine several admixtures and must achieve a very specific and challenging combination of rheological and hydration behavior. Thus, adequate protocols are required to minimize work at the industrial scale, while maximizing the amount of transferrable information that may be gained in the lab. This requires using adequate sample preparation protocols and accepting some degree of admixture adjustment as the mix is scaled up.

### 2.2. Inline Laboratory Reactor for Setting on Demand

The reactor presented in this study is built upon a previous system used for the robotic slip forming of façade mullions installed in the DFAB HOUSE, Empa Dübendorf (Switzerland) [14,15] as well as for casting in the 3D-printed Eggshell formwork with variable cross section, further described in [14]. The system relies on a retarded mix, which was pumped in sequential batches (discontinuously) into a mixer placed above a formwork, in which an accelerator was added. Although the DCS was successfully utilized for these processes [15], the system had several drawbacks that had to be tackled to make it compatible with industrial norms. These can be summarized as: (1) backflow, which was caused by the discontinuity in the pumping, resulting in inconsistency in the rheological properties [16]; (2) size, since the mixing system was designed for a maximum aggregate size of 4 mm, and (3) slow processing rate, which makes the system incompatible with industrial requirements [17].

Consequently, to move towards becoming competitive with standard manufacturing processes used in the industry, the processing system has to be redesigned while also considering the changes in overall mix design required by the industry (see Section 2.3). The major changes in the mixing unit are highlighted in Section 3.3.

### 2.3. Reference Conditions of Industrial Prefabrication

The prefabrication of concrete elements may or may not involve heat curing. Whichever situation, having a concrete that can harden rapidly is interesting because it allows for a faster turn over, especially where heat curing is applied.

The examples given in this paper were developed in collaboration with a Swiss prefabrication company. Specifically, the first case deals with the mass production of high-strength structural columns without heat curing for architecture and infrastructure. Their reference production conditions include steel or wooden formworks. Concrete-mix designs are pre-programmed, and a software controls the weighing and dosage of the different components, before delivering them to a pan mixer. Due to variable seasonal storage conditions, the water content of sand and aggregates may vary from one batch to the other, requiring constant monitoring of the added water to avoid detrimental changes in the final compressive strength.

Typically, 1 m^3^ is mixed for 8 min, after which the concrete is poured into a hopper and transported to the casting hall. Via crane, the hopper is hovered above the formworks and the material is directly poured into traditional formworks (wood or steel) in one large batch. The casting time per column is 1–2 min, depending on the density of the reinforcement. After 8–10 h, the columns are demolded and vertically lifted by a crane into a non-insulated storage hall and positioned horizontally for further curing until being shipped to the customers. To avoid cracking during this step, a compressive strength of at least 15 MPa is required.

The well-established processing logistics and the short casting time make this process efficient. However, for large orders, lead time becomes long as the factory cannot house infinite amounts of formworks. Thus, the use of rapid-setting and hardening concrete presents an appealing solution. The project undertaken here shows that setting on demand would allow rheology to be closely controlled before placing, in order to increase the production rate by 50%, as well as to establish modified processing that would be applicable to the production of concrete elements of non-standard shape by digital casting into weak formworks.

### 2.4. Specific Requirements for Material and Structures

The prefab columns considered in this paper had to be developed in accordance with the following requirements, which were given by the industrial partner:
Final properties
90 MPa after 28 days (strength class C75/90);Fire approved in accordance with standard European building codes;Up to 3 m tall columns with cross sections ranging from 30 × 30 cm to 50 × 50 cm;Good surface quality (smooth, closed and largely uniform);Similar price to the industrial reference.
Processing
Castable in less than 10 min.
Material before acceleration
Good flowability over a sufficient time (e.g., slump flow of 50–60 cm not changing more than 5 cm over 4 h).
Material after acceleration
Reach 15 MPa within 4 h;For ordinary formworks: pumpable for at least 10 min after acceleration;For weak formwork: rapid setting of 2 kPa yield stress 1–2 min after acceleration.

To achieve this, the mix design was revised as follows:Increasing the aggregate size from 4 mm to 8 mm (reducing the paste content);Including short polypropylene fibers (to obtain fire resistance);Changing the cement supplier (cement available at the company in question);Increasing cement replacement by limestone (lower cost, heat release and CO_2_ footprint);Developing a faster and more robust acceleration system (higher industrial performance).

## 3. Materials and Methods

### 3.1. Materials

Our setting-on-demand system includes a set retarder in the concrete base mix, typically a sucrose solution, as well as superplasticizers. Since both these admixtures impact the setting time, their dosages must be carefully adapted to reach the desired setting times and rheological properties, including initial fluidity and flow retention. This optimization may require many iterations, so the best way to operate is typically to carry out preliminary tests in mortar rather than concrete. The same holds for other aspects of the mix design, such as the partial replacement of cement by limestone and/or other SCMs. The characterization of the used materials is presented in Appendix A.

The mix design of the base-retarded concrete is presented in Table 1. The retarded concrete is then mixed with an accelerator. We initially used either suspensions of calcium silicate hydrate or solutions largely based on calcium nitrate. Neither approach was fully satisfactory, as their use implied a considerable addition of water to the mix, making it difficult to avoid segregation upon mixing. Moreover, the setting time and rate of strength development were too slow for the present industrial requirements.

To overcome these limitations, we turned to using suspensions of calcium aluminate cement (CAC) that we had developed for 3D extrusion printing [2,18]. The composition of the CAC paste is given in Table 2. It should be noted that the mix includes a combination of calcium aluminate cement and anhydrite, which avoids the risk of sulfate imbalance in case the dosage has to be increased. It also contains a retarder that prevents it from reacting before it gets mixed with the mortar or concrete. The mix design optimized in the laboratory and used at industrial scale is presented in Table 3.

### 3.2. Test Methods

The first step in adjusting the base mix design is to regulate the dosage of superplasticizer and retarder by also trying to keep the W/C ratio as low as possible, which improves the mechanical properties of the mix while making it sufficiently flowable. This is best done using a combination of spread flow measurements and calorimetry. The first is done using semi-empirical experiments based on slump flow tests according to standard procedures described in DIN EN 12350-5:2019-09 with an open cone 20 cm high with 20 cm diameter at the bottom and 13 cm at the top. The target slump flow was set between 50 and 60 cm over the duration of the open time to ensure enough flowability in the base mix. The sucrose dosage was adjusted to get the desired retardation of the onset of the acceleration period. Results reported in this paper were obtained on mortars using an isothermal calorimeter I-Cal 8000 HPC (Calmetrix, Arlington, MA, USA) at 23 °C.

After the dosage of superplasticizer and retarder has been regulated, compressive strength tests may be performed on the mortars. Provided the same W/C ratio is used as what is planned for the concrete, there should be a good correlation between mortars and concretes. Correlations between heat release and compressive strength as reported by [19,20,21] may be used at this stage, as these may help to adjust the water content, if needed. In addition to these tests, for the mix to be pumpable and self-compacting, a particular range of yield stress and apparent viscosity must be aimed for.

With regard to strength, there are three distinct time frames where target values must be reached, described as periods I, II, and III in Figure 2. Period I starts within the first minutes after the accelerator is added to the retarded mix. It reflects the very early strength gain, a period during which the material transitions from a fluid-like to a solid-like state and where both percolation and strengthening of a solid network within the microstructure take place. We found that this can be well followed by pushing a conical needle into the just-mixed material very slowly. Details are provided by Reiter [22] and Reiter et al. [2]. The method allows yield stresses up to about 200 kPa to be measured.
The second period refers to the demolding strength and is determined by a standard compressive strength measurement (BS EN 12390-3 with a force-controlled mode set at 0.96 kN/s) of 15 cm cubic samples. For this paper, a demolding strength of 15 MPa was targeted at 4 h.

The third period concerns the later strength after 7 and 28 days, which is determined with the same compressive strength measurement (BS EN 12390-3) on 15 cm cubic samples. After casting the cubic formworks, the samples are immediately stored at 20 °C/95%RH and demolded just before the test.

### 3.3. Processing System

The goal of the new digital processing system was to meet industrial norms and assure that it can process the material robustly. Compared to the previous digital casting systems [23], the major changes can be summarized as follows:Change from a discontinuous to a continuous pumping system, which means slow pumping rates can be controlled without starting and stopping the system. This required changing the minimum rotation of the motor of the PFT SWING L Pump from 454 to 46 rotations per minute (RPM).Change the mixing reactor from an open funnel to a closed non-pressurized system (both 1 and 2 were done to prevent backflow).Change the dimensions of the reactor from a single-pin to a double-pin mixing system to increase mixing energy and to have the possibility to mix aggregates up to 8 mm.Change motors from asynchrony motor to server motors in order to:Reduce the weight of the mixing unit.Increase flexibility in the mixing control (back and forth mixing) and individually control impellers.Record the resistance of the motors in voltage (V), which gives an indication of the viscosity of the mixture and can be used in a feedback loop to regulate the mixing rate on the fly during production. This measurement though reflects a high-shear behavior, while concrete placing is a low-shear process. Therefore, this sensor only offers limited insight into yield stress, which is the rheological property of greatest interest, but can nevertheless be used to monitor the consistency of the mixture.Change control software from an open-loop control system to a closed-loop control system.

In Figure 3, a schematic overview of the current setup with the mechanical devices needed is shown. Starting with the concrete conveyer pump, f_c_, the retarded mix is pumped into a reactor with double-mixing pins, S_m_. This operates at a pre-set, but modifiable, rotation rate and includes a measurement of the power V consumed by the motor, as previously mentioned. The accelerating admixture is pumped from the CAC pump, f_cac_, while a peristaltic pump, f_sp_, doses a flow enhancer (superplasticizer) to adjust the fluidity on the fly in the reactor. The accelerated concrete exits the mixer at the top and flows into the formwork.

After several developments on a prototypical level, we settled for the final design depicted in Figure 4. This consists of a vertical chamber designed as a closed (not pressurized) constantly stirred reactor (CSTR). The mixing chamber (a) has a volume of ±3 L. It contains two vertical pins (b) rotating in opposite directions, each with three pins and one paddle stirred by server motors (b1). On each paddle, a rubber is mounted to assure that no material is attached to the wall of the chamber. The spacing between the pins is 3-times larger than the larger aggregate size so the concrete can flow through easily. The retarded concrete is pumped continuously into the CSTR from the bottom (d) and flows out at the top of the mixer (e) into a formwork. The accelerator and flow enhancers are dosed into the chamber through (c). The system allows for a variable flow rate between 1.5 L/min and 3 L/min. These values are based on studies that indicate an optimal residence time combining good mixing with limited thickening of the mix. The control software synchronizes the concrete pump flow rate (f_c_) with the dosing speeds of the peristaltic pump (f_sp_) and the accelerating admixtures pump (f_cac_) to ensure constant dosage of the mix components using a customized software based on [24].

## 4. Material Development and Prototyping

### 4.1. Material

#### 4.1.1. Strength Development

The following section describes the results for the material development using different amounts of accelerator dosages, varying from 10 to 15 w%/cement of CAC, with two superplasticizer dosages in the accelerator paste, 6 and 10 w%/cement. Results also compare deviations based on the flow rate and mixing process. “Offline” refers to experiments conducted without the use of the “inline” digital process described in Section 3.3. Initial strength gains (period I) are shown in Figure 5. Values below 2 kPa are highlighted by the grey band in Figure 5, at the top. They correspond to strengths for which the material cannot sustain itself over more than about 5 cm, which we take as a limit for initial setting. This parameter is most relevant when working with non-standard thin formworks, which are particularly susceptible to the hydraulic pressure in the mix.

Figure 5 shows that increasing the accelerator dosage shortens the initial setting time and increases the rate of early strength gain (larger slope). However, the flow rate does not seem to affect the initial strength gain, suggesting that the mixing quality is unchanged. Apart from this, we note very different behavior in the samples mixed offline. First, they have a longer initial setting time, and second, they are less sensitive to the accelerator dosage, which underlines the importance of the mixing system in such studies.

For period II, Figure 6 Top shows substantial differences between all samples in the first four hours, with the highest strengths obtained with the highest accelerator dosages. In contrast, for period III, Figure 6 Bottom shows that all samples reach very similar strengths by 7 days and probably even earlier. Indeed, while only two values were measured after 2 days, these are very similar. This independence of the long-term strength from the accelerator dosage is consistent with 3D extrusion printing results obtained with the same accelerator paste [5].

In the long term, strength increases more or less by an additional 30%, from 88 MPa at 28 days to values slightly above 110 MPa after four months (Figure 7). Furthermore, Figure 7 shows that there is no significant difference between the mixes prepared inline and offline. Overall, the strengths obtained match the expectations of high-strength concrete for the application considered.

#### 4.1.2. Setting Control

A central aspect of setting on demand is for the concrete and the accelerator to be stable over a long period of time before being combined. Isothermal calorimetry is a useful means to optimize the admixture dosages needed for that. As an example, Figure 8 shows that the retarded mix only begins its acceleration period at about 8 h.

For both accelerated samples, there is a much higher initial heat release that points to a much more active chemical system. Apart from the initial peak that reflects the reaction of the accelerator, the other peak is delayed by higher accelerator content. This corresponds to the hydration of tricalcium silicate and it can be seen to occur much sooner for the accelerator dosage of 10% than for 15%. A priori, this reflects a modification in the aluminate–silicate–sulfate balance, which is also influenced by a slight increase in superplasticizer dosage.

### 4.2. Prototyping

The new mix design and processing unit were tested in two large-scale applications. The first was the high-strength reinforced concrete columns to be cast in a standard wooden formwork, as shown in Figure 9. The second was a non-standard shape concrete column to be cast in a 3D-printed ultra-thin eggshell-type formwork, with minimal reinforcement further described in Section 4.2.2. 

#### 4.2.1. Casting in Standard Formwork

For the standard columns, the advantage of digital casting over adding an accelerator or using a fast cement is that the open time can be better controlled. In particular, if more than one column is to be cast from a single batch, there is virtually no change in concrete workability between the first and last cast column.

In our example, the formwork dimensions were 15 × 15 × 300 cm, which is representative of a prefabricated column on the Swiss market (Figure 9). The rebar cage was 11 × 11 × 300 cm, made of eight ∅20 mm longitudinal rebars and ∅6 mm stirrups spaced 150 mm from each other. The cover layer (distance from steel to formwork) was 25 mm, with respect to the stirrups, and 31 mm, with respect to the rebars. Overall, the reinforcement density was 11.5% by volume. A cross section of such a column, as planned and as produced, can be seen in Figure 10.

Two columns were produced, the first one with 10% CAC with respect to the total cement in the mix and the second one with 15%. These accelerators were continuously added during the digital casting process using the processing and mixing reactor described in Section 3.3. This experiment helped demonstrate that:The element could be demolded after 4 h (having reached 15 MPa).The formwork (Figure 9 Left) with the rebar cage (Figure 9 Middle) was adequately filled.The surface quality is very good (Figure 9 Right), although it still falls short of the SIA norm for exposed concrete.

The fabrication of both columns was successful. They were cast in 20 min with a filling rate of 3 L/min. The first with 10% CAC was demolded after 4 h when 16 MPa had been reached in a room at 23 °C, while the second one, cast with 15% CAC, was demolded after only 3 h, having reached 18 MPa at approximately the same temperature.

#### 4.2.2. Casting in Non-Standard Formwork

For the non-standard shape, polymer extrusion 3D printing was used to produce an ultra-thin formwork [25]. In this case, the accelerators prevent both the build-up of hydrostatic pressure and the high-pH-induced stress cracking [26].

In the present example, by using the DCS, we can capitalize on the above points and use a formwork with only 1.5 mm wall thickness. The formwork was designed as part of a separate study that had shown that different 3D-printed formworks cross sections have different breakage behavior [25]. Thus, the formwork design merged from being a circle (the strongest geometry), decagon, octagon, hexagon and finally to a square (the weakest geometry), forming the 3 m tall formwork (Figure 11). In addition, the changing cross-section geometry of the formwork had different cross-sectional areas, making it necessary to adjust the filling rate during the production, the values for which are shown on the right of Figure 11. In this case, the reinforcement cage consisted of 4 × 6 mm rods with a length of 2.80 m each, and simple metal wires to keep them in position.

As in the previous example, we used the industry mix described in Section 3.1, but only with 10% CAC. As in the previous experiment with the wooden formwork, the CAC paste was continuously added with the digital processing system described in Section 3.3. However, in contrast to the previous case, the filling rate was lower, since the rise of hydrostatic pressure had to be kept in balance, owing to the much thinner wall thickness. As reported in Figure 11, flow rates varied from 1.50 to 3.00 L/min, depending on the cross-sectional geometry. For means of comparison, the filling rate into the standard wooden formwork of the previous example described was a constant of 3 L/min, adding up to a filling time of 20 min, which was possible because the standard formwork would not burst if the hydration was too slow.

Overall, the digital casting of concrete into this thin formwork was successful and finalized after 66 min, with only a minor formwork breakage in the hexagonal part (see Figure 12). The crack occurred because of a temporary lack of CAC paste in a section of the formwork that was already considered weak and susceptible to hydrostatic pressure due to its geometry with more acute angles [25].

Through this application, we were able to demonstrate that DCS is suitable for precisely adjusting the filling rates on the fly during casting, which, as exemplified in this case, may be particularly necessary to locally control the hydrostatic pressure in relation to varying cross sections throughout the formwork (see Figure 13).

It is also important to note that this formwork was shredded and reused without problems for another print. This demonstrates that such formworks can be inscribed in a circular process, a process that has been tested in a separate study, but not yet reported.

### 4.3. Overview of Results from Experiments to Benchmark in Industry

The examples presented in the previous two sub-sections demonstrate that the same concrete mix design and processing unit may be used for mass-produced standard columns, as well as for bespoke structures. This makes it much easier for industry to expand their offer beyond standard structures, also offering bespoke shapes, for which the formwork may be both easily 3D printed and also recycled.

With regard to differences in the processing conditions and outcome, a summary is provided in Table 4. In particular, it reports some of the main characteristics of columns produced at ETH with different amounts of CAC, as compared with standard columns produced at the industry partner using their mix and filling procedure.

## 5. General Discussion

This paper addressed some of the challenges for transferring advanced concrete manufacturing processes from the lab to industry. Specifically, it targeted delivering both a productivity increase for ordinary production and the facilitation of producing bespoke-shaped structures, both with the same process and material mix design.

### 5.1. Materials Issues

The mix design presented was developed with components used for daily production in industry. It has the characteristics of a pumpable mix, which can be accelerated using CAC pastes without causing sedimentation or cold joints during the casting process. It reached 15 MPa after 3–5 h, which enabled its demolding from standard formwork after the same period. Experiments not reported in this paper showed that achieving the same strength takes up to 6 h when temperatures are below 5 °C. This longer required time is not unexpected and remains reasonable. It suggests an energy of activations substantially lower than 50 kJ/mol, which has otherwise been reported for cement hydration [27,28].

We also underline that changes in rheology are an important and delicate issue. In our case, with the same materials both at ETH and at the industrial plant, we nevertheless had differences in initial fluidity and workability retention under the same conditions. This can result from small differences in the cement state (pre-hydration, etc.) and/or mixing energy. In the case presented, dealing with this required increasing the cement content by 12%, with respect to the daily mix design used by the company. However, this was less cement than in the previous set-on-demand mixes used in digital casting [17]. A key element here was an increased use of a limestone filler (50% with respect to cement), which was additionally very beneficial for achieving the levels of self-compaction and pumpability required for digital casting (Figure 14). Additionally, it must be noted that while the reduction in cement was minimal, a material was made with large aggregates that could obtain strength fast and be processed in the inline processing.

Independently of early strength and temperature variations, the selected mix design reaches compressive strengths up to about 90 MPa after 28 days, which complies with the requirements in the industry for high-strength columns. As previously mentioned, a downside of this mix is its higher cement and paste content. In addition to raising the carbon footprint, this also carries an economic cost. Overall, including the CAC cost, this material ends up being approximately 25% more expensive than the typical mix used in the industry.

While the extra material cost can be, in part, compensated by the higher productivity, further changes in the mix design need to be considered. Among the various options, lowering the cement and/or paste content should be prioritized, while still considering that the material should remain pumpable when retarded and self-compacting after acceleration. Reducing the paste content will most certainly be facilitated by broadening the aggregate grading. However, this would call for additional iterations in the inline mixing unit [29]. While this is conceivable and a topic that we are investigating, it requires a fair amount of effort. Faster gains may be obtained by other changes to the mix design, particularly if the type of cement can be changed to reduce the carbon footprint. Indeed, the type of accelerator presented performs very well with low-carbon cements, without compromising final strength [5].

Fire resistance is another important issue, but it could not be tested within the frame of this project. Such tests are cost intensive and, thus, are only done towards the end of a project. However, they are a requirement for certifying the material, so ultimately, they must be conducted to complete the transfer to industry. Nevertheless, as we included a similar number of polypropylene fibers as the reference certified concrete, our proposed mix should a priori be well-positioned to comply with fire-resistance requirements.

### 5.2. Processing

In terms of processing, we presented the conversion of our semi-continuous digital processing system in a continuous and adaptive one, allowing us to react on the fly to changes in material properties, as well as to changes in cross-sectional geometry. As discussed in Section 3.3 this is made possible by our digital control system. The continuous inline mixing chamber eliminated the unwanted backflow and stabilized the processing, thereby offering reliable, though adaptable, casting conditions, both for standard and ultra-thin 3D-printed formworks. To take this a step further, sensors reflecting the quality of the mixing and the rheological properties would be very useful. Clearly, and as could be expected, the torque sensors used up to now only offer limited insight into this process, as they operate at high-shear rates, while concrete workability mainly relates to yield stress, which is best measured at low-shear rates. Nevertheless, a separate study, not reported in this paper, further investigates the subject matter.

### 5.3. Prototypes

For the case of standard columns, the full-scale demonstrator enables production that is 50% faster than standard production. In practice, this is particularly valuable in periods where high throughput is needed. Companies not using heat curing will enjoy faster demolding, while companies using heat curing may resort to additional non-heated formworks to meet high demands without the high capital investment of additional heated formworks.

Beyond this, we showed that it is possible to completely rethink what a formwork can look like. In particular, the option of 3D printing molds on demand in a material that can be easily recycled paves the way for new horizons. Importantly, and in contrast with extrusion printing, this allows steel reinforcement to be easily incorporated, making the process of structural certification much easier.

Despite these achievements, the industry partner still did not adapt the system to their premises, as it is still considered too slow despite the fast demolding. One limiting factor is that, to make the best use of the faster-hardening concrete, it is necessary to establish faster and more flexible production methods. A first step in this direction is to extensively redesign the production pipeline from mixing to casting. Placing formworks on moving rails would also increase flexibility and require only one or a few digital casting systems. Indeed, as illustrated in Figure 15, ordinary production is regulated around having the second cast curing overnight, which is advantageous in terms of work shifts. With a 50% increase in the production rate, the bottom part of the same figure shows that the final casting takes place two hours later than normally and this calls for a different work time distribution.

## 6. Conclusions

The goal of the work presented here was to move our DCS system from an academic environment into an industrial pre-fab plant and to comply with the requirements set by the industry partner. In doing so, we aimed to demonstrate a potential for increased productivity for industry. Beyond this, we wished to stimulate a rethinking of formwork, showing that structurally efficient shapes can be produced using both less concrete and formwork material in the foreseeable future.

In terms of mix design and processing adaptations, we demonstrated that a laboratory-developed process can be upscaled to meet key industry requirements, in terms of strength and productivity. We also presented one of the first processable digital concrete mix designs to include coarse aggregates. Importantly, we also showed that this mix design and processing unit make it possible to produce both standard elements in standard formwork and bespoke structures in ultra-thin formworks.

In terms of cost and a reduction in cement, our mix still includes high amounts of cement and is costly (Appendix A), resulting in a cost that remains slightly above that of the standard mix used in the prefab plant. While further material optimization may reduce this, pure material costs are expected to remain higher. Therefore, while the mix design presented here increases productivity and works with ultra-thin formworks, it is still not recommended for ordinary industrial production, because of both the higher economic cost and the increased CO_2_ footprint. Nevertheless, the method remains promising given the increased productivity and facilitated production of elements with structurally optimized shapes, so our future work lays in the inclusion of larger aggregates and use of low-carbon cement.

In regard to processing, we converted our system from a semi-continuous to a continuous system, which improved robustness and enabled formworks to be removed 50% faster than in normal practice. For ordinary production throughput, for which the current industry installations are optimized, the filling time (20 min) for standard formworks remains a limiting factor for the industry partner. Current research (not reported in this study) focuses on increasing flow rates by scaling up the mixing unit. That said, a change in the infrastructure and logistics within the plant could also be considered, as pointed out in Section 5.2.

Finally, the question of building structures with fewer materials remains open. Although this study showed that we can rethink formwork and thereby, use less bulk material, the main concern of the industry remains productivity. Still, as designers increasingly request non-standard optimized shapes, the industries that can offer that service will conquer this growing market segment.

## Figures and Tables

**Figure 1 materials-15-03468-f001:**
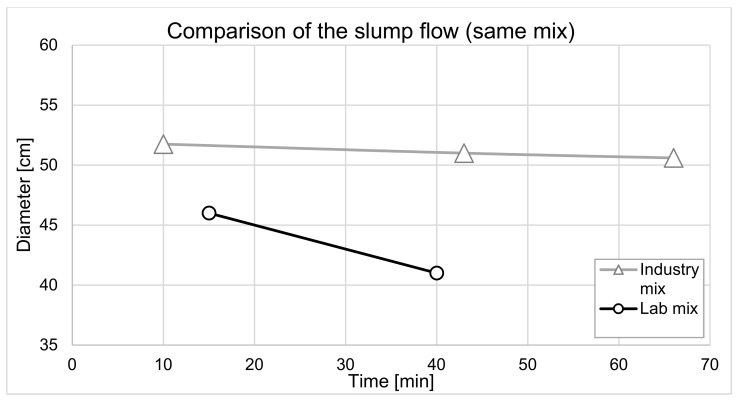
Comparison of the slump flow evolution of a given concrete having the same mix design produced in the industrial and academic laboratory, respectively.

**Figure 2 materials-15-03468-f002:**
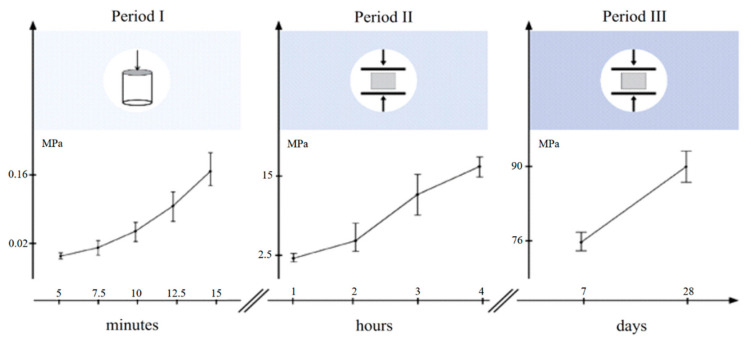
Testing of material strength in 3 different periods: initial strength (**left**) covering the first 15 min after placement, demolding strength (**center**) with the objective of reaching at least 15 MPs at 4 h, late strength (**right**) measured both after 7 days and 28 days.

**Figure 3 materials-15-03468-f003:**
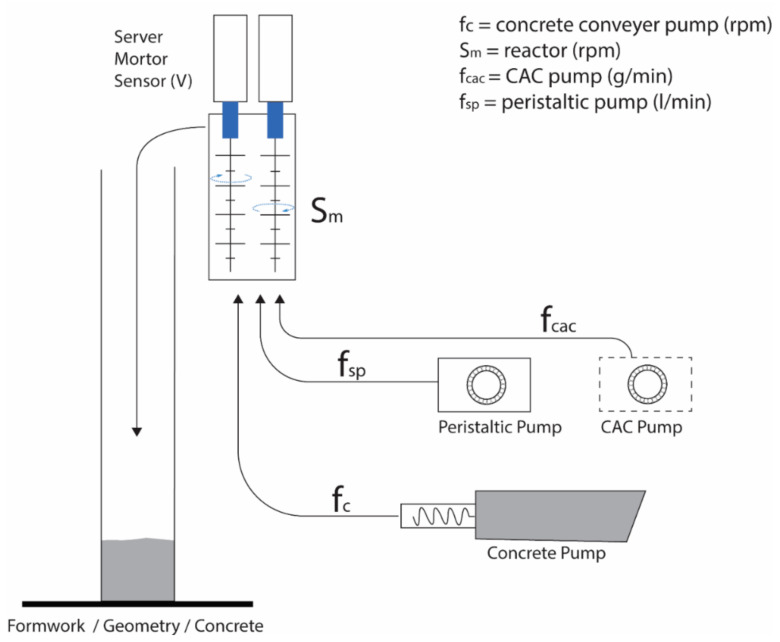
Schematic illustration of pumps and components in the digital casting setup processing system.

**Figure 4 materials-15-03468-f004:**
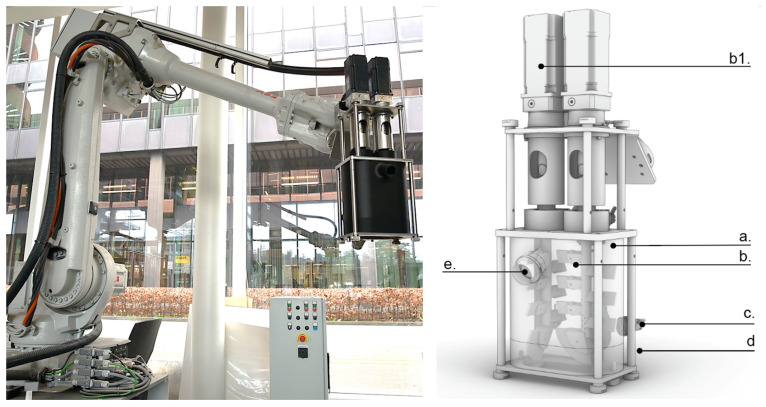
(**left**): Inline mixer mounted on a 6-axis robotic arm; (**right**): schematic overview of components in the inline mixer.

**Figure 5 materials-15-03468-f005:**
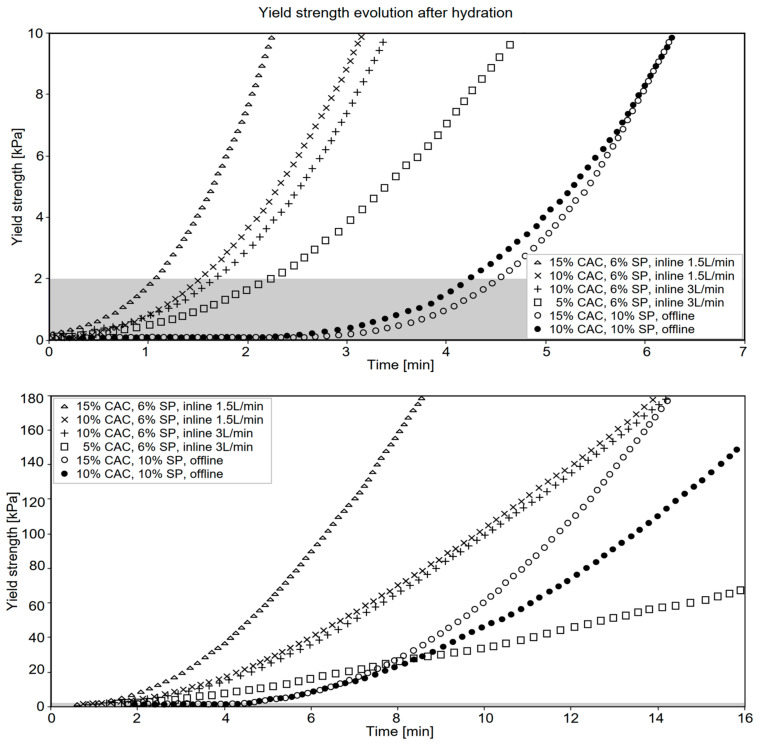
Initial strength development (**period I**). Material strengths after hydration beginning within the first (**top**) 7 min and (**bottom**) 15 min measured by a penetration test.

**Figure 6 materials-15-03468-f006:**
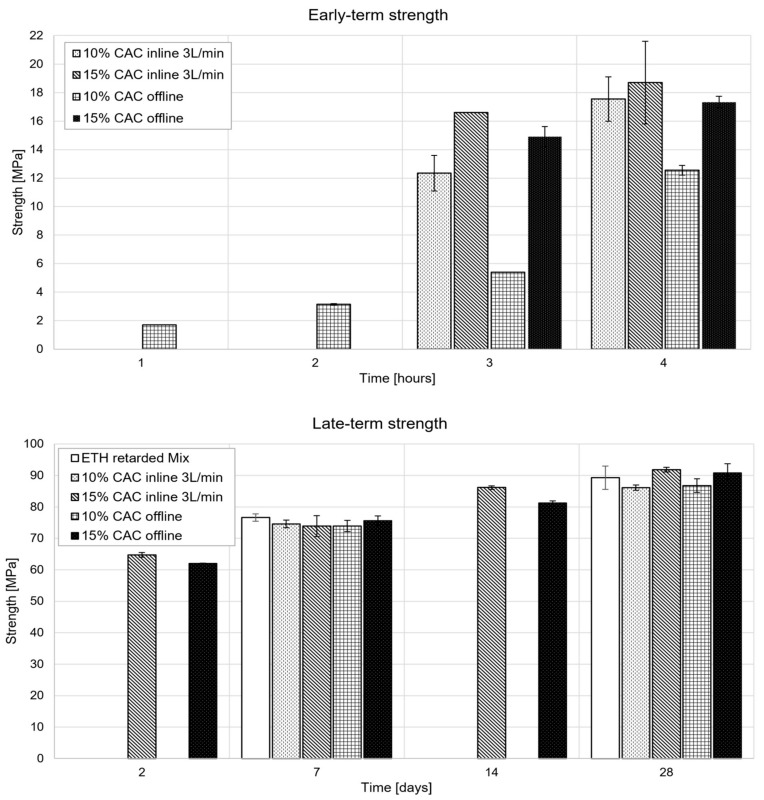
Compressive strength measurements in periods II (**top**) and III (**bottom**). The results present an average of the strengths with corresponding error bars obtained from several samples collected from at least two mixes independently prepared for each category.

**Figure 7 materials-15-03468-f007:**
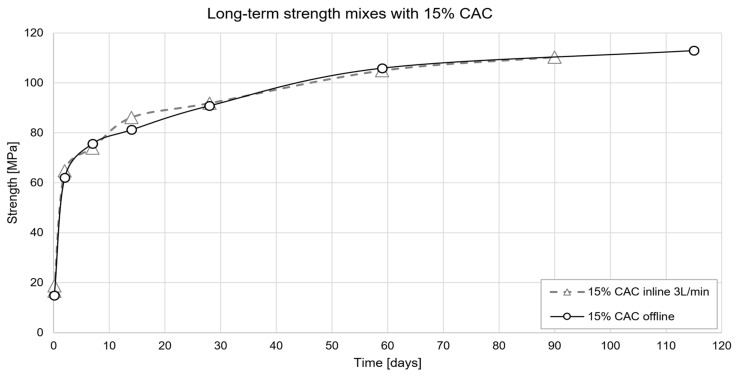
Strength evolution of the accelerated mixes prepared “inline” and “offline”.

**Figure 8 materials-15-03468-f008:**
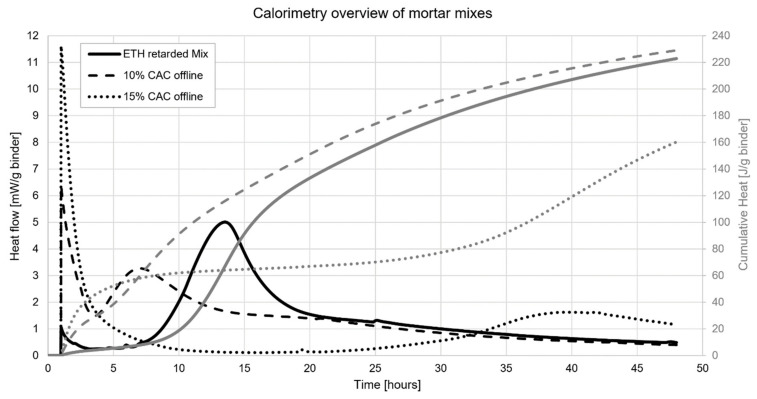
Calorimetry comparison of mortar mixes.

**Figure 9 materials-15-03468-f009:**
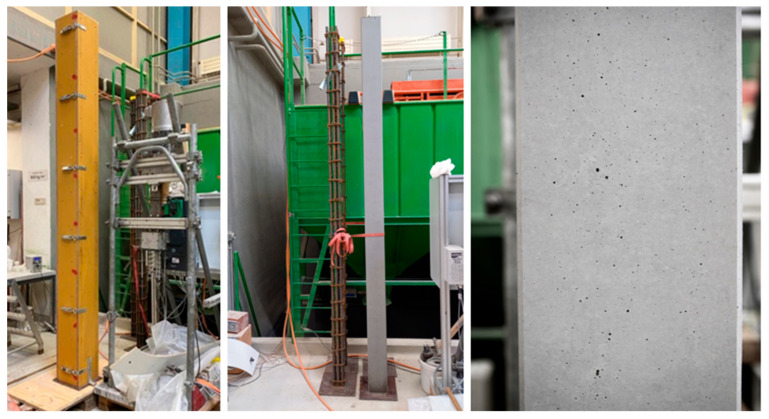
High-strength reinforced concrete column. From left to right: wooden formwork 150 × 150 mm, middle reinforcement cage provided by industry partner, result of casted column.

**Figure 10 materials-15-03468-f010:**
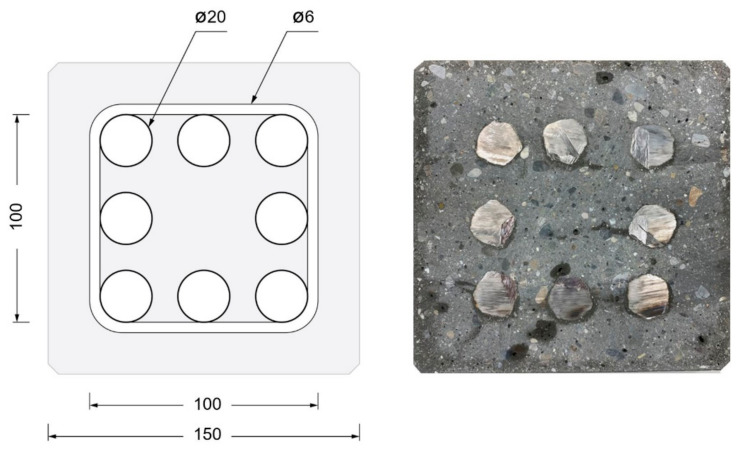
(**Left**): cross-section of a standard formwork with reinforcement. (**Right**): section cut of DCS result in standard formwork using Industry Mix with digital casting. The stirrups are not shown in the picture as the slice made did not cut through one of them.

**Figure 11 materials-15-03468-f011:**
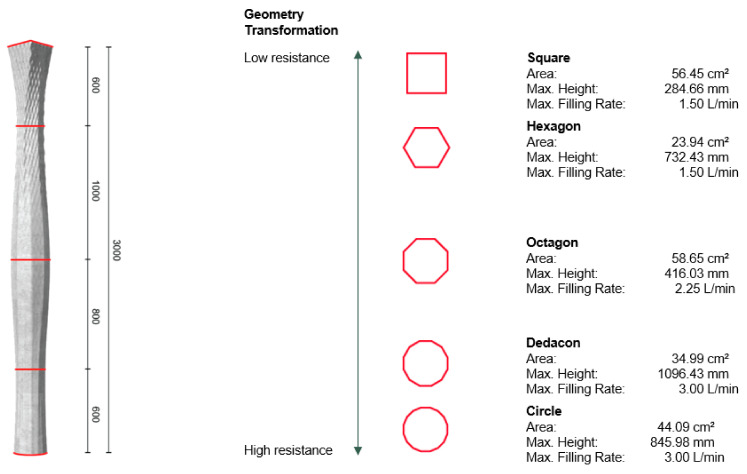
(**Left**) shows the column design intended for the experiment; (**right**) the five lofted geometries with their respective dimensions, volumes and intended filling rates.

**Figure 12 materials-15-03468-f012:**
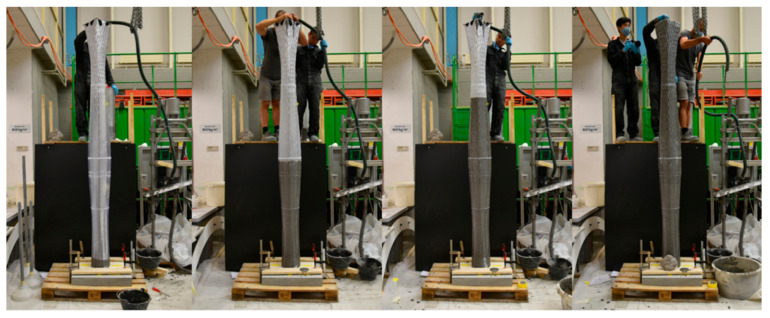
Casting process with DCS with the eggshell formwork.

**Figure 13 materials-15-03468-f013:**
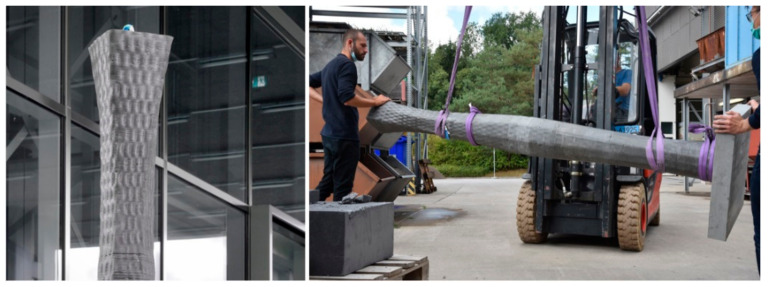
(**Left**) zoom of cast eggshell columns after demolding; (**right**) transport of column.

**Figure 14 materials-15-03468-f014:**
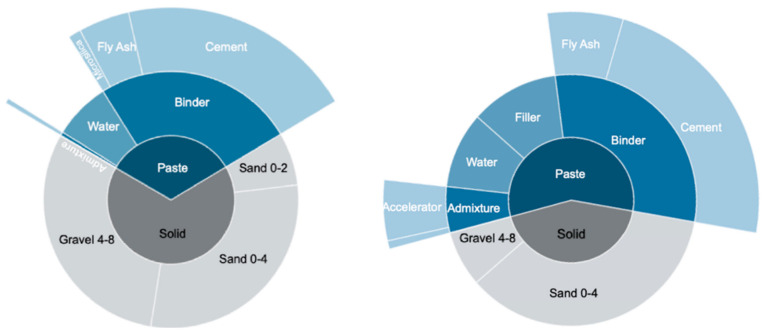
Industry mix composition (**left**) and optimized ETH mix composition (**right**).

**Figure 15 materials-15-03468-f015:**
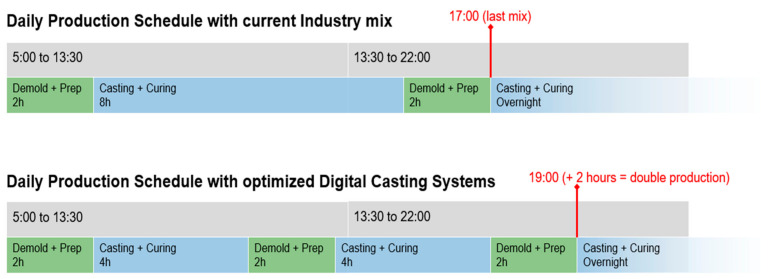
Comparison between the typical production and a production adopting DCS.

**Table 1 materials-15-03468-t001:** Recipe for the ETH retarded (base) mix.

Components: ETH Retarded Mix		Amount [kg/m^3^]	Proportions [w%/Cement]
Binder	Jura CEM I 52.5 R (C)	575.3	-
	Fly Ash Type F	161.1	28.00%
Filler	CaCO_3_ Nekafill 15-KFN	281.9	49.00%
Aggregates	Sand 0–4 mm	849.3	147.63%
	Gravel 4–8 mm	212.3	36.91%
Fibers	PP fibers Wiking M6–18 um	1.6	0.28%
Admixtures	Sucrose solution (solid content 30%)	2.0	0.35%
	Glenium ACE 30 (solid content 30%)	12.7	2.20%
Water	Water added for mixing	231.3	40.22%
	Total W/C = 0.42	241.6	-

**Table 2 materials-15-03468-t002:** Recipe for the accelerator paste.

Components: Accelerator Paste		Amount [kg/m^3^]	Proportions [w%/CAC]
Binder	Calcium aluminate cement (CAC, Ciment Fondu)	1032.6	-
	Anhydrite (Francis Flower)	516.3	50%
Admixtures	PEO-based stabilizer	1.0	0.1%
	Sodium Gluconate (powder, Sigma Aldrich Saint Louis, MO, USA)	1.0	0.1%
Water	Water added for mixing	495.7	48%
	Total W/CAC = 0.48		

**Table 3 materials-15-03468-t003:** Recipe for the final ETH mix with 10% CAC bwc.

Components: ETH Mix with 10% CAC		Amount [kg/m^3^]	Proportions [w%/ Cement]
Binder	Jura CEM I 52.5 R (C)	542.1	-
	Fly Ash Type F	151.8	28.00%
	Calcium aluminate cement (CAC, Ciment Fondu)	54.2	10.00%
	Anhydrite (Francis Flower)	27.1	5.00%
Filler	CaCO_3_ Nekafill 15 -KFN	265.6	49.00%
Aggregates	Sand 0–4 mm	800.3	147.63%
	Gravel 4–8 mm	200.1	36.91%
Fibers	PP fibers Wiking M6–18 um	1.5	0.28%
Admixtures	Sucrose solution (solid content 30%)	1.9	0.35%
	Glenium ACE 30 (solid content 30%)	17.3	3.20%
	PEO-based stabilizer	0.1	0.01%
	Sodium Gluconate (powder, Sigma Aldrich)	0.1	0.01%
Water	Total W/(C + CAC) = 0.43	257.5	-
Properties: ETH Mix with 10% CAC		Amount	Unity
Density		2306	[kg/m^3^]
Paste volume fraction		0.62	-
Solid volume fraction		0.38	-
CAC paste volume to total paste volume		11.7%	-

**Table 4 materials-15-03468-t004:** Overview of results from experiments compared to benchmark in industry.

Material	Casting in Standard Formwork (Benchmark Production)	Casting in Standard Formwork with DCS (10% CAC)(ETH Zürich)	Casting in Standard Formwork with DCS (15% CAC)(ETH Zürich)	Casting Eggshell Formwork with DCS (10% CAC)(ETH Zürich)
Rebars	8 × 20 mm	8 × 20 mm	8 × 20 mm	
Filling time	1–2 min	20 min	20 min	66 min
Filling rate	30 L/min (estimated)	3 L/min	3 L/min	Variable 1.5, 2.25, 3.00 L/min
Curing temp.	23 °C	23 °C	22 °C	
Demolding time	8 h	4 h	3 h	
Compressive Strength after 4 h	--	16 MPa	18 MPa	
Compressive Strength after 28 d	90 Mpa	85 MPa	88 Mpa	
Surface quality	Good	Good	Very good (see Figure 9 Right)

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
