# Peer review of "Additive Digital Casting: From Lab to Industry"

_materials, 2022, doi:10.3390/ma15103468_

Round 1

Reviewer 1 Report

This paper comprehensively described the process from lab to industry. Indeed, the gap between research and practical engineering always limits the application of lab discoveries and this paper provides a good contributor to closing this gap. However, the authors provided a couple of comments that should be addressed before publishing.

  1. The whole manuscript needs some polishing works. The current version needs to be improved by a more concise language.
  2. Lines 70 – 71: On my perspective, beside the sizes, it remains the large differences in mixing environment including temperature, moisture. Please provide more explanations to justify this comment.
  3. Figure 1: I didn't find where this figure was cited. Moreover, it needs more explanations on these two plots. The comparisons have to be clarified which parameter are the controlled and which are the variables; however, I can't tell these things from this figure.
  4. Line 136: “3” should be upper script.
  5. Lines 195 – 197: The paragraph needs to be rearranged.
  6. Conclusion sections are too poor to be published. Please rewrite the entire section with a clear and concise language.

Author Response

Please find the cover letter and answers to reviewers in the attached document

Reviewer 2 Report

Fritschi et al investigated the Digital Casting Systems (DCS) for concrete mixing. The process was first demonstrated at the laboratory level, then upscaled to the industry level.

The quantum of the experimental works is sufficient and the results show some interesting outcomes.

In the introduction section, clearly mention the exact innovation of the current study.

The manuscript is poorly presented. The methodology section is very descriptive and confusing. They should focus on the objectives and goals carefully.

Line No. 75: Error in reference citing.

Did Figure 1 data collect by the authors themselves? Otherwise, provide a citation and make ensure that standard permission has been taken from their publisher(s).

Subsection 2.2 needs a major improvement. It seems like the authors have developed the reactor on the basis of previously reported articles, with some changes. 
The drawbacks or limitations can be confined to a couple of lines with giving reference citations. 
They should highlight what changes were adopted in the new reactor design.

Line No. 100: Correct the superscripts for 70 cm2 and 140 cm2. Do a similar change in Line 106.

Line No. 117: Make all units uniform throughout the manuscript. For example, either you can write 1cm or 1 cm.

Line No. 136. Correct the unit. 1m3!

Subsection 2.4: Were the design requirements as per any Building Standards? If yes, then mention it.

Line No. 231: Error in referencing.

Line No. 243: As per above is confusing. Instead, write the specific methods.

Line No. 271, 282: Error in referencing.

Line No. 332: mention Figure No.

Also, I can see a lot of errors in referencing and figure citations. The manuscript seems to submit in a hurry mode. Kindly take a relook of the whole paper and do the necessary changes.

Author Response

the answers are provided in the document attached 

Reviewer 3 Report

In this regard, the presented paper covers Digital Casting Systems (DCS), whereby the hydration of self-compacting concrete is controlled on the fly during production to enable the use of standard and cheap thin formwork without suffering from hydrostatic pressure. This is an interesting study, well written by the author. Therefore, there are some issues needed to be clarified before it can be accepted for publication in Materials:

  1. The abstract should be refined. Research significance, main content and conclusions should all be reflected.
  2. The author is needed to check the abbreviations and symbols in the manuscript. Pls. use [1–7] instead of [1]–[7]. Line 231. Error! Reference source not?
  3. The author should give the necessary physical and chemical properties in the material section.
  4. Some sentences in the introduction are too long, the author can simplify them.
  5. Tables 1-3. What is the reason of choosing those W/C ratios and the CAC? Please explain it in the manuscript.
  6. Test Methods. The authors tested the rheological properties, but lack the necessary description, pls refer to recent publications (Ceramics International, 2022, 48(9), 12884–12896).
  7. The author should conduct a more in-depth and detailed analysis of the manuscript.

Author Response

(The authors gave the same response as above.)

Round 2

Reviewer 2 Report

The authors have provided a supplementary file for detailed explanations. Also, They have included most of the recommendations made by reviewers. So, the manuscript can be considered for the next stage process.